# Characterization of Canine Otitis Externa *Pseudomonas aeruginosa* Isolates and Their Sensitivities to Different Essential Oils

**DOI:** 10.3390/ani15060826

**Published:** 2025-03-13

**Authors:** Anita Seres-Steinbach, Brigitta Schneider-Patkó, Ákos Jerzsele, Adrienn Mercedesz Veres, Ágnes Sonnevend, Krisztián Bányai, György Schneider

**Affiliations:** 1Department of Medical Microbiology and Immunology, Medical School, University of Pécs, H-7624 Pécs, Hungary; seres-steinbach.anita@edu.pte.hu (A.S.-S.); patko.brigitta@pte.hu (B.S.-P.); pal.agnes@pte.hu (Á.S.); 2Department of Pharmacology and Toxicology, University of Veterinary Medicine, H-1078 Budapest, Hungary; jerzsele.akos@univet.hu (Á.J.); veres.a.mercedesz@univet.hu (A.M.V.); bkrota@hotmail.com (K.B.); 3National Laboratory of Infectious Animal Diseases, Antimicrobial Resistance, Veterinary Public Health and Food Chain Safety, University of Veterinary Medicine, H-1078 Budapest, Hungary; 4HUN-REN Veterinary Medical Research Institute, H-1143 Budapest, Hungary

**Keywords:** *Pseudomonas aeruginosa*, otitis externa, genetic relation, antibiogram, essential oil, antibiofilm activity, alternative therapy

## Abstract

Otitis externa is a very common infectious inflammatory disease in dogs that affects the external ear canal. This infection can be caused by different bacteria, among which *Pseudomonas aeruginosa* is the greatest challenge for practitioners, as it can form biofilms and is highly resistant to antibiotic treatments, resulting in frequent therapeutic failure. To find alternatives to antibiotics for its topical treatment, we tested the capacity of essential oils to kill and degrade the mature biofilm of 40 distinct *P. aeruginosa* isolated from dogs’ otitis externa. We identified six essential oils: cinnamon (bark and leaf), bay, rosemary, thyme, and bitter orange with ability to kill all the tested *P. aeruginosa* isolates and destroy established biofilms, a crucial maintenance factor for chronic infection. With this study, we demonstrate that certain essential oils could be used as a reliable topical alternative in the treatment of otitis externa in dogs.

## 1. Introduction

Otitis externa is a very common infectious inflammatory disease in dogs presenting to veterinary clinics worldwide [1,2]. This condition affects the external ear canal, including the ear pinna. A high percentage of these cases is associated with the presence of *Pseudomonas aeruginosa*, a Gram-negative opportunistic pathogenic bacterium that survives in a wide range of environments [3] and prefers to colonize the ear canals of different mammals. The topical treatment of this infection mostly includes aminoglycosides (mainly gentamicin and tobramycin), polymyxin-B, and fluoroquinolones, but, recently, therapeutic failures have been reported [2] due to the presence of antibiotic resistance among the causative agents. The eradication of *P. aeruginosa* has become increasingly difficult due to its remarkable capacity to resist antibiotics. Restricted outer-membrane permeability, efflux systems, and antibiotic-inactivating enzymes are crucial mechanisms of the intrinsic antibiotic resistance of *P. aeruginosa* by which this bacterium can spoil or completely neutralize the antimicrobial effects of fluoroquinolones, beta lactams, aminoglycosides, and polymyxins [4]. In addition to these intrinsic factors, it can also gain additional antibiotic resistance mechanisms through mutational changes or the acquisition of resistance genes via horizontal gene transfer [5]. As a result, multidrug-resistant strains can appear and emerge.

One reason for therapeutic failures is that although the currently used veterinary antibiotics, such as aminoglycosides (gentamicin and neomycin), fluoroquinolones (ciprofloxacin, enrofloxacin, marbofloxacin, and orbifloxacin), and polymyxins (colistin sulfate and polymyxin B), are effective, the biofilms produced by these isolates inhibit their uptake and, thus, inhibit, or at least impair, their effects. This self-produced matrix of extracellular polymeric substances (EPSs), like proteins, LPS, alginate, and metabolites, has the capacity to protect the residing bacteria from various environmental effects and also from antibiotics [6,7].

According to recent surveys, 90.6% of isolates of *P. aeruginosa* from dogs and 86.4% of isolates from cats produce a biofilm [4]. The rates of therapeutic failure, which are influenced by intrinsic antibiotic resistance and biofilm formation, vary widely among studies performed in different geographical regions, but recently it has been estimated that nearly 24% of treatments of otitis externa cases in dogs are unsuccessful and that this is more pronounced if *P. aeruginosa* is the causative agent, as in these cases, 80–90% of cases are untreatable [8]. This situation can establish an epidemiological situation in which multidrug-resistant *P. aeruginosa* is transmitted from the pet to the owner and, depending on the virulence attributes of the infectious agent, it can cause human infections in the ears, eyes, or in the respiratory or urinary tracts. It is therefore important to find alternative ways to efficiently treat *P. aeruginosa* infections.

In terms of gaining control of this veterinary and epidemiological situation, essential oils (EOs) might offer a solution. Several EOs have been described to have antibacterial activities. Their compositions are diverse and their mechanisms of action are complex. It has become evident that they not only contribute to the rapid lysis of the bacterial cell membrane, thereby undermining the osmotic stability of the cell, but that certain compounds are able to directly influence molecular mechanisms in the cell, such as synthetic processes (RNA, DNA, protein) or enzymatic actions [9,10], and influence biofilm formation; thus they may be suitable alternatives to traditional antibiotics.

In the present study, our aim was to reveal the antibacterial efficacies of 57 EOs on 40 distinct canine otitis external *P. aeruginosa* isolates. The therapeutic potentials of these EOs were assessed by revealing their effects on planktonic bacterial cells and also on mature biofilms. All experiments were performed at 30 °C and 37 °C in order to reveal any possible temperature dependence of the effects within the critical application range. Based on our results, *L. nobilis*, *R. officinalis,* and *T. vulgaris* were the most effective EOs, as none of the *P. aeruginosa* isolates showed sensitivity against them.

## 2. Materials and Methods

### 2.1. Bacterium Isolation and Identification (MALDI-TOF)

*P. aeruginosa* strains were isolated from dogs with clinical signs of otitis externa. Samples inoculated directly onto MacConkey agar were cultured at 37 °C. Identification was performed by Gram staining, microscopic examination, and growth on selective cetrimide agar (Sharlau Chemie, Sentmenat, Spain), colony morphology, pigment production, and oxidase positivity. Identification was confirmed by matrix-assisted laser desorption/ionization time-of-flight mass spectrometry (MALDI-TOF MS) (Bruker, Billerica, MA, USA). Each isolate was preserved at −80 °C in Mueller–Hinton broth supplemented with 20% glycerol and subcultured on 5% sheep blood agar plates prior to the experiments.

### 2.2. Antibiotic Sensitivity Testing

Antibiotic testing performed according to EUCAST v.14 [11] was followed, but with slight modifications. Bacterial suspensions were synchronized to McFarland 0.5 and plated on blood agar plates. Six-millimeter diameter antibiotic discs (Bio-Rad Laboratories, Hercules, CA, USA) were placed on the surface of the obtained bacterial lawn and incubated at 37 °C overnight (O/N). The size of the zones of inhibition was measured and expressed in millimeters.

The following antibiotics were used to determine the antibiogram: Piperacillin + tazobactam 30/6 μg (PTZ 36), imipenem 10 μg (IPM 10), meropenem 10 μg (MEM 10), ceftolozan + tazobactam 30/10 μg (CLT 40), ceftazidime 10 μg (CZD 10), cefepime 30 μg (FEP 30), ceftazidime + avibactam 10/4 μg (CZA 14), gentamicin 10 μg (GMN10), amikacin 30 μg (AKN 30), ciprofloxacin 5 μg (CIP 5), tobramycin 10 μg (TMN 10), levofloxacin 5 μg (LVX 5), and trimethoprim + sulfamethoxazole 1.25/23.75 μg (STX 25).

### 2.3. Pulsed Field Gel Electrophoresis

Genotyping based on the macrorestriction patterns was performed as described by Selim et al. [12] with slight modifications. Individual isolates were inoculated onto 5% sheep blood agar and incubated overnight at 37 °C. Bacterial suspensions were prepared in ethylenediaminetetraacetic acid (EDTA) saline buffer (75 mmol/L NaCl and 25 mmol/L EDTA, pH 7.5) with optical density adjusted to 3.5–4.0 McFarland. These suspensions (300 µL) were mixed with 700 µL plug agarose (1% Lonza SeaPlaque GTG agarose in sterile dH_2_O), and plugs were poured and allowed to solidify for 30 min at room temperature. The agarose plug was incubated for 5 h at 37 C in 10 mL of lysis buffer (6 mmol/L Tris-HCl (pH 7.6), 0.1 mol/L EDTA, 1 mol/L NaCl, 0.4% sodium deoxycholate, 0.5% sodium lauryl-sarcosine, and 1 mg/mL lysozyme). Next, the lysis buffer was replaced with 5 mL of Proteinase K buffer (1% sodium lauryl sarcosine, 0.5 mol/L EDTA (pH 9) with 0.125 mL proteinase K (Invitrogen) (20 mg/mL)), and the tubes were incubated overnight at 54 °C The plugs were then washed five times for 20 min at 50 °C with 10 mL of TrisEDTA buffer (10 mmol/L Tris-HCl (pH 8) and 1 mmol/L EDTA) in a shaking water bath, and stored in 5 mL TE buffer at 4 °C.

Plugs were digested with 30 units of SpeI-HF restriction enzyme (New England Biolabs, Ipswich, MA, USA) using a *Salmonella* serotype Braenderup strain (H9812) as a universal size standard. The H9812 plugs were digested with 30 units of XbaI restriction enzyme (New England Biolabs) and the plugs with the restriction enzymes were incubated at 37 °C overnight.

Restricted fragments were separated in 1.2% PFGE Running Agarose Gel (Merck, Darmstadt, Germany) in 0.5× TBE buffer at 14 °C on a CHEF Mapper XA system (Bio-Rad Laboratories, Hercules, CA, USA), by using an initial switch time, 2 s; a final switch time of 40 s; with linear ramping for 20 h run time; included angle of 120°; 6.0 V/cm.

After electrophoresis, the gel was stained with ethidium bromide after electrophoresis and the bands were visualized using the GEL DOC XR gel documentation system (Bio-Rad Laboratories, Hercules, CA, USA). Similarities of macrorestriction patterns were calculated using GelCompare II (Bionumerics, Sint-Marten-Latem, Belgium), with >80% similarity being considered as genetically related.

### 2.4. Antibacterial Testing of the Essential Oils—Drop Plate Method 

Five isolates with characteristically different antibiotic susceptibility profiles were used to pre-screen the antibacterial efficacy of the 57 EOs (A.G Industries, Noida, India) by using the classical drop plate method. On the basis of the results obtained, the 6 most potent essential oils were selected for testing.

For the tests, 100 µL suspensions (OD600 = 0.2) were spread on Mueller–Hinton agar plates. After drying, 5 μL of 2.5% and 5% essential oils (diluted in 1% Tween-20) were dropped onto the bacterial lawns and incubated for 24 h at 30 °C and 37 °C, respectively. Inhibition zones were measured on the following day and expressed in millimeters.

### 2.5. Growth Kinetics

Growth kinetics experiments were carried out to investigate possible differences in growth characteristics among the isolates. The tests were carried out in two different media, LB and MH, at two different temperatures, 30 °C and 37 °C. One hundred and ninety-eight mL of MH and LB were pipetted into the wells of non-adhesive 96-well tissue culture plates, to which 2 µL of the corresponding bacterial suspensions (OD620 = 0.2) were added. Measurements were performed in a multimode reader (Allsheng, Hangzhou, China) over a 24 h period, with optical densities measured every 15 min at the wavelength of 630 nm. A 5 s shaking step was incorporated before each measurement. The acquired data were converted into Excel files and the curves were plotted. Monitoring was performed in triplicate (3 wells/isolate).

### 2.6. Biofilm Assay

The traditional crystal violet binding assay was performed [13] at 30 °C and 37 °C to determine the biofilm-forming capacities of the 40 otitis externa *P. aeruginosa* isolates. Each test was performed in 3 different 96-well cell culture polystyrene plates (Sarstedt, Nümbrecht, Germany), such as plates for adhesive cells (83.3924), for complex adherent cells (83.3924.300), and for suspension cells (83.3924.500).

For the procedure, optical densities of 17 h bacterial suspensions were synchronized to OD600 = 0.2 in LB medium. Ten microliters from these bacterial suspensions were transferred to the wells of 96-well polystyrene microtiter plates containing 190 μL broth. The plates were incubated at 30 °C and 37 °C for 24 h. After incubation, the culture medium containing the planktonic cells was carefully removed from each well and washed three times with PBS. The biofilm was fixed with 200 μL 2% formalin-PBS solution (28.6 mL, 35% formalin + 471.4 mL PBS) for 2 min, then the formalin was removed and the plates were dried at 37 °C for 2 h. The so fixed cells were stained with 0.13% crystal violet (14.29 mL 35% formalin + 234.41 mL PBS + 1.302 mL 96% ethanol + 0.325 g crystal violet) for 20 min at room temperature and then washed three times with PBS. The stained biofilm layers were solubilized with 200 µL of 1% sodium dodecyl sulphate (SDS) dissolved in 50% ethanol (96%) and 50% PBS. After 2 h, the absorbance of the solubilized crystal violet solutions was measured at 630 nm in a microplate reader (AMT-100T, Allsheng, Hangzhou, China) and plotted.

### 2.7. Biofilm Degradation Assay

The biofilm inhibitory capacities of the tested essential oils were determined based on the previously described biofilm crystal violet assay with slight modifications. Accordingly, after 1 day of incubation, the wells were carefully washed 3× with PBS, and then 180 µL of LB medium and 20 uL of 2.5% and 5% essential oils were added, reaching final concentrations of 0.25 and 0.5%, respectively. The wells were incubated for a further 24 h at 37 °C, after which the planktonic cells were carefully removed by washing; then they were fixed, stained, and detected, as previously described (Section 2.6).

### 2.8. Statistical Analyses

The balloon plot was made using the freely accessible SRplot (URL: https://www.bioinformatics.com.cn/srplot, Accessed on 12 November 2024). Linear regression analysis was used to identify factors influencing biofilm formation (JASP 0.18.3.0).

One-way ANOVA was used to determine whether there was a significant difference between the individual essential oils and their concentrations, as well as the temperatures applied, in the biofilm-forming ability of different isolates. Post hoc analysis was then used to detect differences using Tukey’s test. In both cases, *p* < 0.05 was the significance threshold.

## 3. Results

### 3.1. Isolation and Identification of P. aeruginosa Otitis Externa Isolates

A total of 40 different *P. aeruginosa* were selected in frame of this project. According to the samples, they were all dominant species from the individual infections, and their MALDI scores were between 2.00 and 2.20.

### 3.2. Susceptibility of the P. aeruginosa Isolates

The antibiotic resistance profiles of the isolates showed that they were all susceptible to piperacillin + tazobactam, imipenem (PTZ), meropenem (MEM), cefepime (FEP), and the combination of ceftazidime and avibactam (CZA). In contrast, they were all resistant to the combination of trimethoprim and sulfamethoxazole (STX), but this was used as a control as it is not usually tested in case of Pseudomonas (Table 1).

Resistance to ceftolozane + tazobactam (CLT), gentamicin (GMN), amikacin (AKN), ciprofloxacin (CIP), tobramycin (TMN), and levofloxacin (LVX) was observed in some isolates.

Isolate 7 was the only isolate that was resistant to two antibiotics, while isolate 8 showed resistance to a new generation antibiotic combination such as ceftolozane + tazobactam (CLT) (Figure 1 and Table 1).

### 3.3. Pulsed-Field Gel Electrophoresis

On the basis of the criteria used in the macrorestriction analysis, the following isolates were more closely related to each other: 4–22, 17–19, 2–31, 7–15, 1–13 (Figure 2).

### 3.4. Antimicrobial Effects of Essential Oils—Drop Plate Method

In the pre-screening tests, of the 57 essential oils, 27 essential oils had no effect on *Pseudomonas aeruginosa* when tested at the 2.5% concentration, while 24 essential oils were ineffective at a higher (5%) concentration (Figure 3 and Figure 4). Examples of such EOs were *Piper nigrum*, *Elettaria cardamomum*, *Anethum sempervirens*, *Santalum album*, etc.

Among the effective EOs, six, i.e., *Laurus nobilis* (bay), *Cinnamomum zeylanicum* (bark), *Cinnamomum zeylanicum* (leaf), *Citrus aurantium* (bitter orange), *Rosmarinus officinalis* (rosemary), and *Thymus vulgaris* (thyme), were selected for further testing as they showed convincing antibacterial activities against the 40 isolates. *L. nobilis*, *R. officinalis,* and *T. vulgaris* inhibited all isolates to some extent, and none of them showed any resistance (Figure 5).

In general, cinnamon (*Cinnamomum zeylanicum*) was the most effective, regardless of whether the essential oil was distilled from the bark or the leaves. In the case of bark, inhibition zones ranged from 5 to 7.5 mm when 2.5% essential oil was used, while 7–8 mm could be measured when 5% essential oil was used. In the case of cinnamon essential oil extracted from leaves, the inhibition zone was 5.5–8 mm at the 2.5% concentration, while it was 6–7.5 mm at the higher concentration. Bitter orange (*Citrus aurantium)* showed a similarly convincing effect, with inhibition zone sizes ranging from 4.3 to 7.2 mm depending on concentration applied.

The efficacy of noble laurel (*Laurus nobilis*) was more variable, with no inhibition zone observed for isolate 8, and 3–7 mm for others.

Rosemary (*Rosmarinus officinalis*) essential oil had a good inhibitory effect on 38 isolates (2–6 mm), but there was no measurable zone of inhibition for isolates 8 and 27.

Thyme (*Thymus vulgaris*) showed results similar to rosemary (2–6 mm), but no zone of inhibition was observed for isolates 5 and 38.

### 3.5. Growth Kinetics

Some characteristic differences among the growth kinetic patterns of the tested *P. aeruginosa* isolates were revealed when performed in different media and temperatures.

In the case of the isolates studied, the combination of BHI and 30 °C proved to be the most effective in promoting their growth (Figure 6a–f).

Isolate 33 (Figure 6e) showed the most different growth characteristics, as the 24 h incubation time was not sufficient for this bacterium to reach the stationary phase. This characteristic was somewhat similar in the case of isolate 37 (Figure 6f), but in this latter case the progression of the kinetic curve was more moderate, and by the 10th and 13th hours of the incubation, the intensity of the logarithmic phase was broken and the ramp-up of the curve became more moderate. However, by this time, other isolates had already reached the stationary phase (8; Figure 6b) and were already beginning to enter the declination phase (4, 10, 13; Figure 6a,c,d). This characteristic feature was dependent on the medium used.

### 3.6. Biofilm Assay

Regarding the cumulative results of the biofilm assay, *P. aeruginosa* can be divided into two groups. The biofilm-forming ability of the first group is influenced only by temperature and medium quality. In contrast, the other group consisted of those isolates whose biofilm-forming ability was not only influenced by temperature and medium quality, but also by the adhesion properties of the applied tissue culture plate.

When comparing the biofilm-forming abilities of the isolates, it was found that 30 °C was much more conducive to biofilm formation (Figure 7). In addition, isolate 37 showed an increased ability to form biofilm at both temperatures, in each medium and in all tissue culture plates, and had excellent OD values at 630 nm compared to the other isolates.

At 30 °C, isolate 23 had the worst biofilm-forming ability, as it only reached OD: 0.024–0.082 in MH-II, while in LB it showed values of 0.06–0.165, depending on the tissue culture plate used. Its biofilm-forming ability was strongly influenced by the applied nutrient liquid and plate.

Isolate 7 had the worst biofilm-forming ability at 37 °C. It is also characteristic of this isolate that its ability to form biofilms is influenced by the plate and the nutrient fluid. In the case of MH II, OD was 0.01–0.06, while its optical density in LB broth was OD: 0.049–0.11.

Isolates 4 and 13 formed a biofilm in a balanced manner at both 30 °C and 37 °C, independent of the nutrient fluid and plate, but had a higher optical density at 30 °C.

The ability of isolate 10 to form biofilm was influenced by the surface of the plate used and the culture medium. Contrary to the previous strains, this strain formed a higher biofilm at 37 °C compared to 30 °C. The biofilm-forming ability of isolate 33 was temperature-dependent and was slightly affected by the used tissue culture plate and the liquid medium.

Statistical analysis of the biofilm formation assay using the linear regression method confirmed that temperature significantly affected the biofilm formation ability of most isolates (*p* < 0.01) (Figure 8). At 30 °C, *P. aeruginosa* formed 25% more biofilm than at 37 °C (Figure 8a). In addition, the surface quality also had a drastic effect on the ability of the isolates to form biofilm. 

Regarding the carrier medium, the statistical analysis confirmed that biofilm formation was 9% more efficient on LB than on MHII.

### 3.7. Biofilm-Degrading Capacities of EOs

The comparative analysis of the biofilm-disrupting potential of the five essential oils studied revealed their potential to eliminate the already established matrix structure. In general, cinnamon proved to be the most effective by having the ability to destroy the biofilm structure under the conditions studied even when applied at a lower concentration. A closer look revealed significant differences between the EOs depending on concentration, exposure time, and, certainly, the EO itself (Figure 9). In each case, *p* < 0.01. Therefore, we performed the post hoc analysis to see the significant values more precisely.

In the post hoc analysis, we examined the significance between the essential oils used, their concentrations, and the incubation times.

In the case of *Cinnamonum zeylanicum*, there was no significant difference in biofilm removal after the first hour when it was used at the final concentrations of 0.25% or 0.5%, as both concentrations were very effective in breaking down the biofilm. However, a significant difference was measured between the 1 h and 24 h values, as an even more effective biofilm removal was observed after 24 h.

In the case of laurel (*Laurus nobilis*), there was no significant difference between the degrading effects of the 0.25% and 0.5% concentrations in the first hour, and there was no significant difference after 1 day of incubation using the 0.25% concentration. However, in the case of the 0.5% concentration, a significant difference in optical values could be measured as early as the first hour.

The biofilm-disrupting ability of petitgrain (*Citrus aurantium*) was similar at 0.25% and 0.5% in the first hour, so there was no significant difference between the two concentrations, but as the exposure time increased from 1 h to 1 day, then the biofilm-disrupting effect also increased.

In the case of rosemary (*Rosmarinus officinalis*), a similar phenomenon was observed as it did not show a drastic biofilm-degrading effect in the first hour of incubation, whether it was used at 0.25 or 0.5% concentration. However, as the incubation time increased (1 day), a significant difference was observed between the eradication effects of 0.25% and 0.5% rosemary.

In general, no significant differences were observed between the 1 and 24 h results for the 0.5% concentrations.

In the case of thyme (*Thymus vulgaris*), there was no significant difference between the first hour of biofilm eradication with 0.25 or 0.5% essential oil, and no significant difference between the first and 24th hour results of 0.25% essential oil. However, after 24 h, a significant difference was observed between the 0.25 and 0.5% essential oils.

When comparing the biofilm-disrupting effects of the individual essential oils, a significant difference could be measured. The biofilm destruction effect of the 0.25% *Cinnamonum zeylanicum* after the first hour showed a significantly better effect when compared to either the 0.25% or the 0.5% concentrations of bay, petitgrain, rosemary, or thyme.

It is worth mentioning that when the post hoc analysis was extended to the concrete isolates, the efficacy of the essential oils varied by strain (Appendix A). On the other hand, in several cases, the efficacy of the essential oils also depended on temperatures and other factors (Appendix A).

## 4. Discussion

Canine otitis externa is one of the most common canine diseases worldwide [14]. Due to its characteristic ability to form biofilms and to become resistant to several antibiotics, *P. aeruginosa* was considered the most persistent pathogen of this infection and was the focus of this study.

As the spread of antibiotic resistance affects the success rate of therapy for otitis externa in dogs, alternative methods must be considered as potential therapeutics. In this study, we investigated the potential use of EOs, focusing on antibacterial properties and biofilm-disrupting abilities. The differences, revealed by PFGE, as the primary genotyping method [15], confirmed that the 40 isolates collected for our study from different cases of canine otitis externa were different (Figure 1). The low rate (17.5%, 7/40) of antibiotic resistance of our isolates was moderate when compared with international data, as in other studies performed in Brazil, Turkey, Japan, and Spain [14,15,16,17], this rate was around 30.7–39.53% or more. There could be more reasons for the differences, but the use of antibiotics in the different countries could be one them, which can actually induce different rates of antibiotic resistance. Other reasons could be the type of the investigated population studied, and methodologies used could also cause discrepancies.

From a therapeutic point of view, resistances to gentamicin and fluoroquinolones, two antibiotics that are typically the first line of treatments for otitis externa treatment in dogs, associated with Pseudomonas, are important. Of the 40 isolates tested, 2 (5%, 2/40) were resistant to gentamicin (Table 1), which is consistent with international data, which are otherwise quite broad (3–43%) [15,18,19,20,21,22]. High levels of resistance to fluoroquinolones (i.e., CIP) have also been reported in canine *P. aeruginosa* isolates (24–86%) [15]. In our case, resistance to ciprofloxacin was low (2.5%), meaning that only 1 isolate out of 40 was resistant to this group of antibiotics. In contrast, recent articles have reported high levels of resistance to fluoroquinolones (i.e., CIP) among isolates derived from various canine infections [23], which is an alarming situation because ciprofloxacin is still a potential therapeutic option for the treatment of many human diseases, similar to its relative, enrofloxacin, in the veterinary practice [23]. According to a survey, in Japan alone, between 0.81 and 0.91 tons of fluoroquinolones were used each year (from 2013 to 2018) for the treatment of otitis externa in dogs and cats [15].

Thus, in contrast to the relatively low occurrence of antimicrobial resistance among our isolates (Table 1), international data highlight the association between the extensive use of antimicrobial agents in dogs and the increasing antimicrobial resistance in canine otitis externa isolates of *P. aeruginosa* [19]. A high rate of antimicrobial resistance in canine isolates not only leads to the spread of antimicrobial resistant isolates among dogs [24], which reduces the success rate of veterinary therapy, but can also pose a potential threat to public health because of the likelihood that such dogs will serve as vehicles for the spread of antimicrobial-resistant bacteria and elements [24,25,26] to humans.

If we are to reduce this epidemiological burden, we need to reduce the use of antibiotics where possible. One option for this is the use of essential-oil-based therapeutics. The demonstration of the antibacterial properties of several essential oils from the 57 studied on different *P. aeruginosa* otitis externa isolates (Figure 3 and Figure 4) strongly supports this approach.

Concentration dependence of antibacterial activity is an important issue, and our results may provide guidance for final concentrations of ointments and liquid formulations to be tested in the future. The demonstration that the 2.5% concentration of EOs is effective in eliminating bacteria from surfaces and is also effective in biofilm removal (Figure 8) somewhat determines that this concentration range needs to be considered in future applications. The biofilm eradication potential is crucial for the effective treatment of otitis externa, as cases are very often caused by biofilm-forming bacterial pathogens, where the biofilm matrix, a self-produced extracellular polymeric substance (EPS), consisting of proteins, LPS, alginate, and different metabolites, has the ability to protect the resident bacteria from various environmental influences and to induce tolerance and resistance to antibiotics [6,7]. This is important, because 40–95% of *P. aeruginosa* isolates from otitis externa produce biofilm [1]. The strong antibiofilm effect of cinnamon and petitgrain (Figure 9) suggests that certain EOs have the ability to disrupt the complex matrix structure and kill the hiding *P. aeruginosa* isolates. Currently, ionophores and antimicrobial adjuvants are being investigated to facilitate this process with varying degrees of success [27].

It is not yet known which compounds of EOs are responsible for the biofilm-degrading effect, but for future compound-based research, cinnamon could be an ideal candidate to clarify this, as this EO had the best performance under different exposures and conditions when compared to bay, petitgrain, rosemary, and thyme (Figure 8). The main constituent of the cinnamon oil is cinnamic aldehyde, together with cinnamyl acetic ester and some cinnamic acid and eugenol [28]. Among these, the antibiofilm effect of cinnamic aldehyde has recently been demonstrated in the case of *Staphylococcus aureus*, and *Malassezia* spp. [29,30], two microorganism that are also associated with otitis externa in dogs, but also in the case of the Gram-negative *E. coli* [31].

Cinnamon is not only interesting from an antibacterial point of view, but may also contribute to the complex management of otitis externa in dogs, as one of its compounds, eugenol, has an anti-inflammatory effect [32]. This offers the possibility of a complex therapy as it not only kills bacteria but can also reduce the effects of the inflammation that accompanies the symptoms.

## 5. Conclusions

The emergence of resistance, and especially the presence of multidrug-resistant *P. aeruginosa* isolates, sometimes makes the therapy of canine otitis externa cases a real challenge. Here, we suggest that the use of essential-oil-based therapeutics as alternative medicines could solve this problem. Some of them not only have an antibacterial property, but also have the ability to suppress biofilm formation and destroy mature biofilm, an important aspect influencing bacterial survival and, thus, successful therapy. As biofilm formation impairs the effects of antibiotics, and as certain EOs have the ability to inhibit or destroy biofilms, and also based on our results, a potential topic is the investigation of synergistic effects of the currently used antibiotics and certain EOs. Our results are promising, but future studies need to clarify the mode of action of candidate oils and address practical issues such as dosage forms and applications.

## Figures and Tables

**Figure 1 animals-15-00826-f001:**
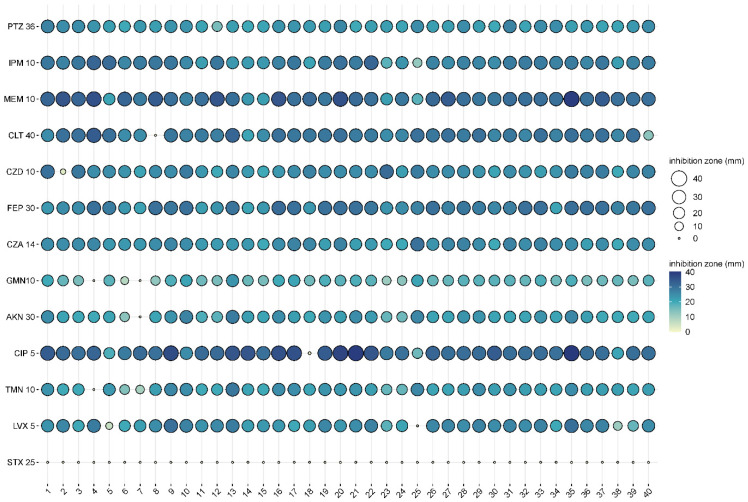
Antibiograms of the 40 different canine otitis externa *P. aeruginosa* isolates (1-40: first column with bold), represented in balloon plot. Abbreviations of the used antibiotics: piperacillin + tazobactam 30/6 µg (PTZ 36), imipenem 10 µg (IPM 10), meropenem 10 µg (MEM 10), ceftolozane + tazobactam 30/10 µg (CLT 40), ceftazidime 10 µg (CZD 10), cefepime 30 µg (FEP 30), ceftazidime + avibactam 10/4 µg (CZA 14), gentamicin 10 µg (GMN10), amikacin 30 µg (AKN 30), ciprofloxacin 5 µg (CIP 5), tobramycin 10 µg (TMN 10), levofloxacin 5 µg (LVX 5), and trimethoprim + sulfamethoxazole 1.25/23.75 µg (STX 25).

**Figure 2 animals-15-00826-f002:**
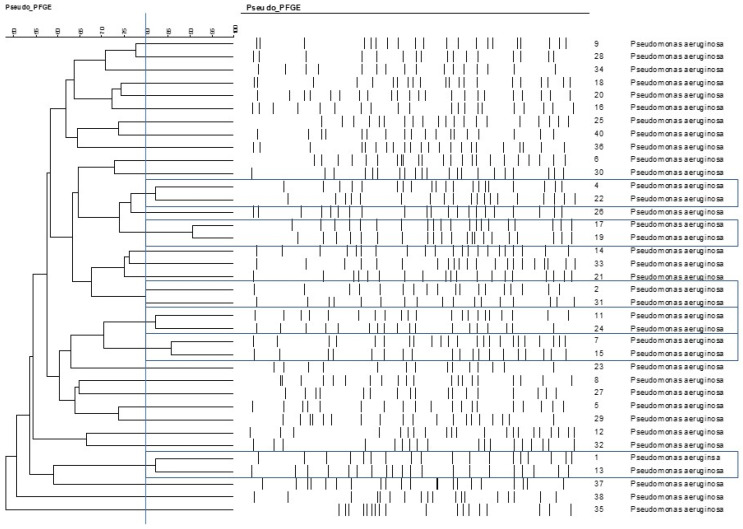
Relationships among the otitis externa *Pseudomonas aeruginosa* isolates, represented on cladogram. The cut-off point for different PFGE types was 80% similarity, labelled with the bue vertical line. Based on that criteria closely related isolate pairs are framed with blue lines.

**Figure 3 animals-15-00826-f003:**
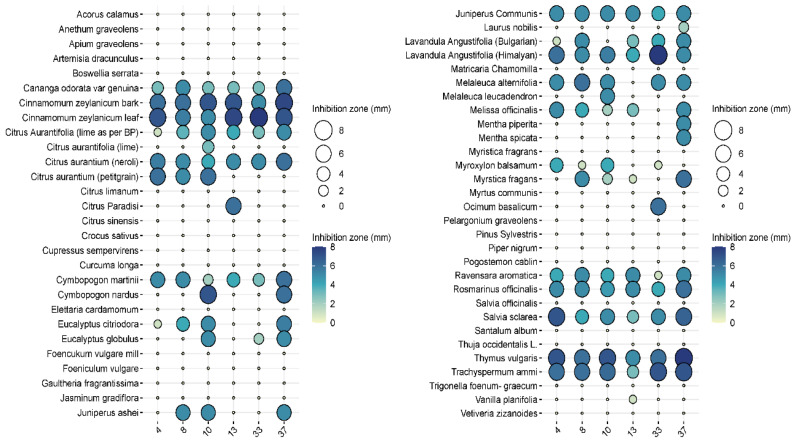
Results of the antimicrobial tests performed with the 57 essential oil solutions at 2.5%. The diameters of the inhibition zones of 10 μL drops are expressed in mm and shown as circles on the balloon plots.

**Figure 4 animals-15-00826-f004:**
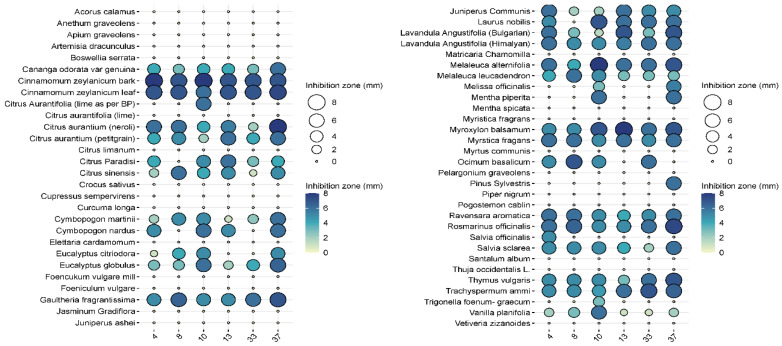
Results of the antimicrobial tests performed with the 57 essential oil solutions at 5%. Diameters of the inhibition zones of 10 μL drops are expressed in mm and shown as circles on the balloon plots.

**Figure 5 animals-15-00826-f005:**
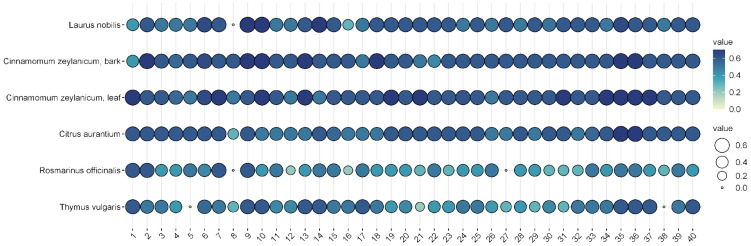
Antibacterial efficacy of the 5% suspensions of the six selected EOs on the 40 different otitis externa *P. aeruginosa* isolates. The plots represent the sizes of the inhibition zones of the bacterial lawns on the 40 different *P. aeruginosa* isolates. The colors and circles represent the diameters of the inhibition zones in mm.

**Figure 6 animals-15-00826-f006:**
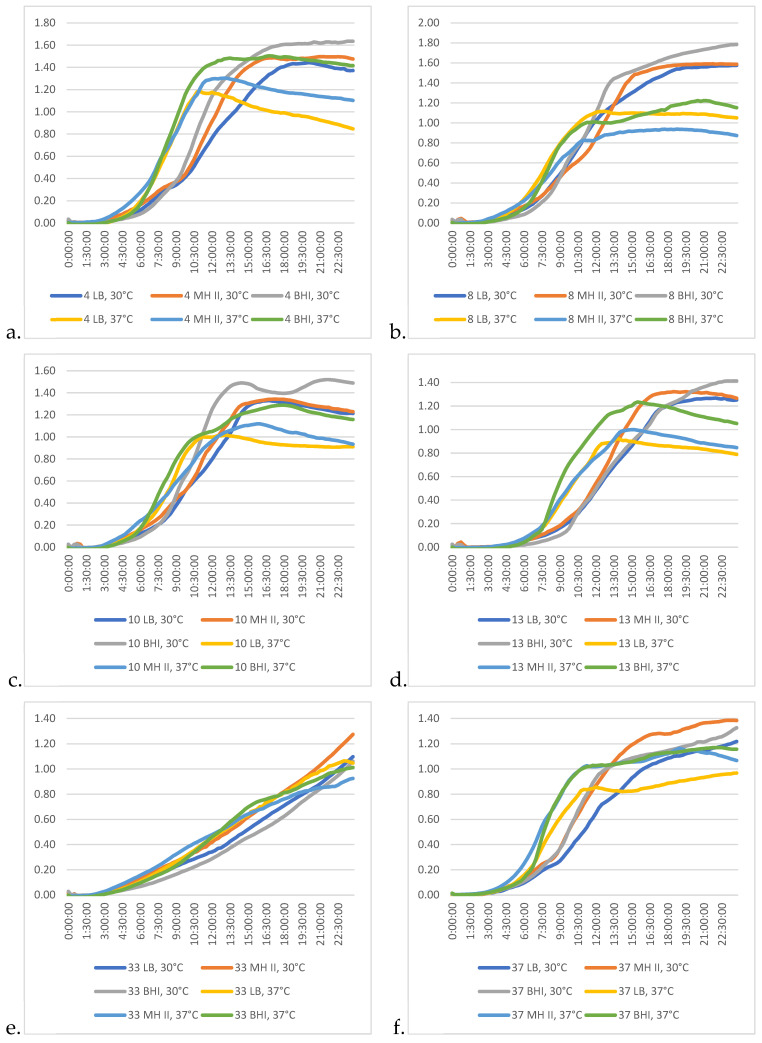
Differences among the growth kinetic patterns of the studied *P. aeruginosa* isolates at different temperatures and in different media. (**a**) Isolate 4; (**b**) isolate 8, (**c**) isolate 10; (**d**) isolate 13; (**e**) isolate 33; (**f**) isolate 37.

**Figure 7 animals-15-00826-f007:**
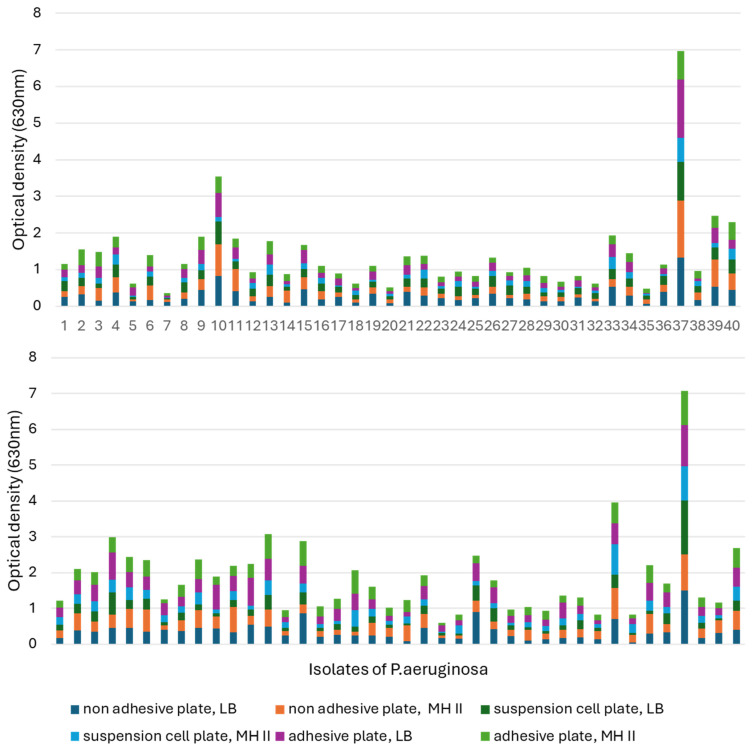
Comparison of the biofilm formation capacity of the 40 otitis externa *P. aeruginosa* isolates at (**top**) 37 °C and (**bottom**) 30 °C. Comparisons were performed in LB and MHII media by using 96-well tissue culture plates for adherent, non-adherent, and suspension cells.

**Figure 8 animals-15-00826-f008:**
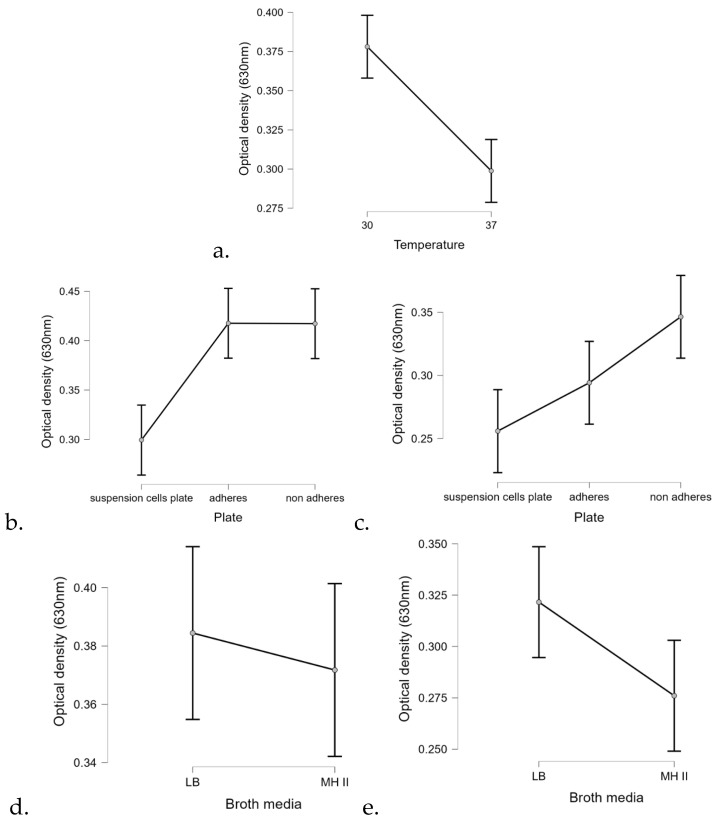
Marginal effects of (**a**) temperature, plate quality at different temperatures (**b**) 30 °C and (**c**) at 37 °C, and the applied broth media at different temperatures (**d**) at 30 °C and (**e**) at 37 °C on biofilm formation of the 40 tested *P. aeruginosa* isolates.

**Figure 9 animals-15-00826-f009:**
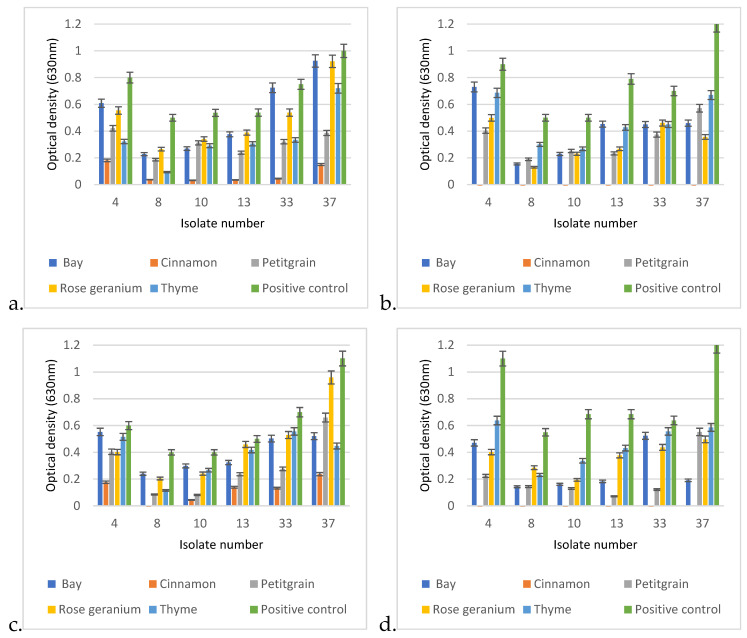
Biofilm degradation capacities of the 5 tested EOs under different exposure conditions on established biofilms of otitis externa *P. aeruginosa* isolates 4, 8, 10, 13, 33, and 37. The degradation conditions were given in the following order: applied EO concentration, exposure time, applied temperature. Based on this, the graphs represent the following exposure conditions: (**a**) 0.25%, 1 h, 30 °C; (**b**) 0.25%, 24 h, 30 °C; (**c**) 0.5%, 1 h, 30 °C; (**d**) 0.5%, 24 h, 30 °C; (**e**) 0.25%1 h, 37 °C; (**f**) 0.25%, 24 h, 37 °C; (**g**) 0.5%, 1 h, 37 °C; (**h**) 0.5%, 24 h, 37 °C.

**Table 1 animals-15-00826-t001:** Exact inhibition zone values of the tested antibiotic discs on the lawn of the 40 different canine otitis externa *P. aeruginosa* isolates. Abbreviations of the antibiotics: piperacillin + tazobactam (PTZ 36), imipenem (IPM 10), meropenem (MEM 10), ceftolozane + tazobactam (CLT 40), ceftazidime (CZD 10), cefepime (FEP 30), ceftazidime + avibactam (CZA 14), gentamicin (GMN 10), amikacin (AKN 30), ciprofloxacin (CIP 5), tobramycin (TMN 10), levofloxacin (LVX 5), and trimethoprim + sulfamethoxazole (STX 25).

*P. aeruginosa* Isolate	PTZ	IPM	MEM	CLT	CZD	FEP	CZA	GMN	AKN	CIP	TMN	LVX 5	STX
**1**	27.0	29.0	32.0	25.0	30.0	23.0	25.0	20.0	25.0	34.0	24.0	24.0	0.0
**2**	25.0	27.0	35.0	30.0	2.8	25.0	25.0	17.0	21.0	31.0	20.0	26.0	0.0
**3**	25.0	29.0	32.0	30.0	29.0	25.0	25.0	15.0	20.0	30.0	20.0	20.0	0.0
**4**	24.0	32.0	36.0	34.0	25.0	30.0	26.0	0.0	20.0	32.0	0.0	28.0	0.0
**5**	22.0	31.0	21.0	30.0	25.0	28.0	23.0	18.0	21.0	19.0	23.0	6.0	0.0
**6**	20.0	26.0	32.0	25.0	24.0	23.0	23.0	8.0	13.0	28.0	14.0	20.0	0.0
**7**	23.0	28.0	28.0	25.0	21.0	21.5	23.0	0.0	0.0	31.0	10.0	22.0	0.0
**8**	22.0	29.0	34.0	0.0	25.0	30.0	25.0	12.5	21.0	30.0	22.0	25.0	0.0
**9**	24.0	29.0	30.0	29.0	25.0	29.0	26.0	19.0	24.0	37.0	26.0	30.0	0.0
**10**	23.5	25.5	27.0	27.0	28.0	30.0	25.0	21.0	27.0	25.0	25.0	27.0	0.0
**11**	21.0	22.0	30.0	28.0	23.0	22.0	22.5	16.0	19.0	31.0	20.0	24.0	0.0
**12**	15.0	29.0	35.0	27.5	20.0	24.0	22.5	15.0	18.0	31.0	20.0	25.5	0.0
**13**	22.0	23.0	30.5	32.0	26.0	29.5	23.0	24.0	28.0	35.0	29.0	28.0	0.0
**14**	20.0	22.5	23.0	21.0	22.5	21.0	23.0	16.5	22.0	35.0	21.0	22.0	0.0
**15**	21.0	23.0	22.0	25.0	20.0	23.0	20.0	15.0	21.0	30.0	20.0	23.0	0.0
**16**	24.0	27.0	35.0	30.0	23.0	31.0	25.0	20.0	25.0	36.0	25.0	23.0	0.0
**17**	23.5	29.0	30.0	29.0	25.0	30.0	27.0	21.0	22.0	36.0	25.0	20.0	0.0
**18**	22.5	21.0	31.0	29.0	27.0	23.0	26.0	16.0	20.0	0.2	21.0	22.0	0.0
**19**	22.0	28.0	29.0	29.0	25.0	30.0	22.0	19.0	24.0	34.0	24.0	26.0	0.0
**20**	26.0	30.0	36.0	30.0	25.0	30.0	27.0	18.0	22.0	38.0	23.0	23.0	0.0
**21**	20.0	28.0	31.0	28.0	25.0	30.0	22.0	19.0	25.0	40.0	24.0	25.0	0.0
**22**	22.0	32.0	30.0	28.0	25.0	29.0	22.5	18.0	22.5	35.0	22.0	27.0	0.0
**23**	22.0	18.0	22.0	25.0	31.0	24.0	20.0	11.0	17.0	28.0	17.0	18.0	0.0
**24**	21.0	24.0	29.0	26.0	22.0	25.0	20.0	13.0	16.0	30.0	17.0	21.0	0.0
**25**	25.0	12.0	18.0	30.0	28.0	27.0	30.0	21.0	26.0	16.0	26.0	0.0	0.0
**26**	22.0	27.0	31.5	29.0	25.0	30.0	24.0	17.0	21.0	32.0	21.0	26.0	0.0
**27**	22.0	25.0	34.0	30.0	25.0	27.0	26.0	17.0	21.0	30.5	21.0	26.0	0.0
**28**	25.0	27.0	30.2	28.0	25.0	29.0	27.0	18.0	24.0	30.6	22.0	25.0	0.0
**29**	22.0	26.0	29.0	30.0	24.0	28.0	25.0	18.0	24.0	30.7	25.0	27.0	0.0
**30**	20.0	25.0	29.0	26.0	21.0	26.0	20.0	18.0	21.0	34.0	22.0	25.0	0.0
**31**	27.0	28.0	30.6	29.0	25.0	27.5	27.0	18.0	21.0	30.5	24.0	26.5	0.0
**32**	21.0	28.0	30.2	26.0	24.0	30.0	25.0	17.0	21.0	30.3	22.0	25.0	0.0
**33**	25.0	28.0	30.2	30.0	22.0	30.0	27.0	19.0	24.0	30.6	22.0	27.0	0.0
**34**	25.0	27.0	30.2	27.0	24.0	21.0	25.0	16.0	21.0	30.0	21.0	20.0	0.0
**35**	22.0	28.0	40.0	30.2	28.0	30.3	25.0	19.0	26.0	40.2	25.0	30.5	0.0
**36**	23.0	26.0	30.0	28.0	27.0	30.0	22.0	15.0	22.0	30.5	22.0	26.0	0.0
**37**	21.0	29.0	34.0	30.0	28.0	30.1	25.0	16.0	20.0	30.5	22.0	27.0	0.0
**38**	21.0	22.0	30.0	26.0	20.0	27.0	20.0	17.0	20.0	21.0	22.0	11.0	0.0
**39**	23.0	27.0	30.0	30.0	26.0	29.0	25.0	15.0	20.0	30.5	21.0	17.0	0.0
**40**	25.0	28.0	30.1	13.0	27.0	30.0	25.0	18.0	21.0	30.5	22.0	25.0	0.0

## Data Availability

All data are provided in the text and the Appendix A.

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
