# Peer review of "Characterization of Canine Otitis Externa Pseudomonas aeruginosa Isolates and Their Sensitivities to Different Essential Oils"

_animals, 2025, doi:10.3390/ani15060826_

Round 1
Reviewer 1 Report
Comments and Suggestions for Authors
Comments:
This study is important because it offers a promising alternative to conventional antibiotics in the treatment of canine otitis externa, reducing the risk of antimicrobial resistance development. It also highlights the efficacy of essential oils in degrading bacterial biofilm, which can significantly improve the success of veterinary therapy.
In the following, I will provide a general analysis and comments for each section of the paper.
Abstract
The summary is clear, well-structured, and provides an overview of the study. However, you should emphasize the study's objective more clearly in the first sentence.
The keywords are relevant, but for better academic visibility, you could include terms such as "alternative therapy" or "antibiofilm activity."
Introduction
In the introduction, you should make a slight adjustment to the objective to provide clarity, impact, and a precise direction for the study.
Materials and Methods
After analyzing this section, several areas were identified where sentences could be more concisely reformulated, avoiding unnecessary repetition.
Example:
Lines 94–96 should have the manufacturer details removed, as they are not essential here, and the two sentences should be combined into a single, more concise statement.
Lines 98–100: The first sentence should be revised by excluding "Using the classical method" as it is redundant, and "disk diffusion" since it is already understood in the Kirby-Bauer method.
Lines 105–106 should be reformulated by eliminating the repetition of "the plates were incubated", as the subject is already clear from the previous sentence.
Lines 133–134, should be revised similarly by removing redundant details while maintaining clarity.
The "Materials and Methods" section is well-documented; however, it could be slightly condensed by reformulating sentences to eliminate repetitions and redundant phrasing. The suggested revisions maintain clarity without losing essential information.
Results
For better reader comprehension, when presenting the results of Table 1, abbreviations should be used in the text. Additionally, the transitions before the analysis of isolates 7 and 8 should be improved. This would help the reader understand why these specific isolates are analyzed in detail and how their results relate to the data in the table.
The section "Antimicrobial Effects of Essential Oils" should be revised to eliminate repetitions and to rephrase the results regarding the effect of essential oils. Additionally, you should provide information about the inhibition zones in the case of bitter orange essential oil.
At the end of the "Biofilm Assay" section, after Figure 6, you presented the numerical results, but it is not clear why they are important and how they influence the treatment of the infection. You should establish a clearer connection between the experimental results and their clinical implications.
Discussion
Lines 432-434: You mentioned a low resistance rate compared to international data, but you did not provide an explanation for this. How do you explain this difference compared to similar studies? You should include a possible interpretation, considering factors such as the type of population studied, antibiotic usage, geographical variations, or applied methodologies.
Lines 455-456: You should create a smoother transition between antibiotic use and the proposed solution (essential oils). Currently, the connection between these two aspects is not sufficiently clear, and a more detailed explanation could strengthen the justification for using essential oils as an alternative or adjunct in treatment.
Lines 460-463: You stated that out of the 57 essential oils tested, only 6 were effective, but you did not explain whether their effectiveness varied depending on concentration, exposure time, or specific conditions. It would be helpful to clarify these aspects to provide a more comprehensive interpretation of the results and facilitate comparisons with similar studies.
Lines 465-469: You stated that therapy is influenced by biofilm, but you did not explain how biofilm affects conventional treatments. You should provide a clear explanation of the mechanisms by which biofilm reduces antibiotic effectiveness, such as physical protection of bacteria, metabolic changes, or the development of antimicrobial tolerance and resistance. Additionally, it would be useful to emphasize why biofilm degradation is critical for treatment success.
Lines 480-482: You mentioned the antibacterial and anti-inflammatory effects of eugenol, but you did not explain how these properties could contribute to the treatment of external otitis. You should establish a clear connection between eugenol’s bioactive properties and its potential clinical role, explaining the mechanisms through which it may inhibit bacterial growth, reduce inflammation, and alleviate symptoms. This clarification would help in understanding how eugenol could be used as an adjunct or alternative treatment.
Conclusions
The conclusions should be reformulated to clarify the following aspects:
The necessity of antibiotic alternatives – Highlighting the increasing antimicrobial resistance and the urgent need to identify alternative solutions, such as essential oils. The relevance of this study should be emphasized in the context of finding sustainable and effective options.
The link between Pseudomonas aeruginosa, zoonoses, and the One-Health concept – Explaining how this bacterium impacts both human and veterinary health, thus justifying the importance of an integrated approach based on One-Health principles.
Feasibility of replacing antibiotics with essential oils – Mentioning the clinical and experimental context that supports the use of essential oils as alternatives or adjuncts in treatment. Additionally, discussing potential limitations and advantages, including compound stability, bioavailability, and administration methods.
The importance of the study and future research directions – Emphasizing the impact of this study in opening new research pathways regarding effective combinations of essential oils, their mechanisms of action, and applicability in biofilm-associated infections.
This clarification will strengthen the conclusions and provide a more scientifically grounded framework for the use of essential oils in combating P. aeruginosa-related infections.
In general, the references included in the manuscript are relevant and well-chosen for the context of the study. However, to enhance the scientific foundation and the up-to-date nature of the bibliography, I recommend adding recent studies (2020-2024) that address the use of essential oils in veterinary medicine and their effectiveness against Pseudomonas aeruginosa.
Comments on the Quality of English LanguageSometimes, present and past tenses are incorrectly used within the same sentence.
Additionally, there are errors in article usage (a/an/the), and in some places, essential articles are missing.
Minor spelling mistakes are also present, (e.g. "infleuncing" instead of "influencing").
Some sentences are too long, leading to word order mistakes.
There are verb agreement errors and overly technical phrasing, lacking clear transitions between ideas.
Recommendation:
A professional proofreading or a careful review by a fluent English speaker would significantly improve clarity and readability.
Author Response
Reviewer 1
This study is important because it offers a promising alternative to conventional antibiotics in the treatment of canine otitis externa, reducing the risk of antimicrobial resistance development. It also highlights the efficacy of essential oils in degrading bacterial biofilm, which can significantly improve the success of veterinary therapy.
In the following, I will provide a general analysis and comments for each section of the paper.
- Dear Reviewer, thank You for your positive opinion!
Abstract
The summary is clear, well-structured, and provides an overview of the study. However, you should emphasize the study's objective more clearly in the first sentence.
- we modified the first and second sentences of abstract.
The keywords are relevant, but for better academic visibility, you could include terms such as "alternative therapy" or "antibiofilm activity."
- we added these terms now
Introduction
In the introduction, you should make a slight adjustment to the objective to provide clarity, impact, and a precise direction for the study.
- we made some adjustments, mostly focusing ont he last paragraph of this section.
Materials and Methods
After analyzing this section, several areas were identified where sentences could be more concisely reformulated, avoiding unnecessary repetition.
- We reformulated this sections and methodological repeats were minimized.
Example:
Lines 94–96 should have the manufacturer details removed, as they are not essential here, and the two sentences should be combined into a single, more concise statement.
- done
Lines 98–100: The first sentence should be revised by excluding "Using the classical method" as it is redundant, and "disk diffusion" since it is already understood in the Kirby-Bauer method.
- we reformulated the sentence
Lines 105–106 should be reformulated by eliminating the repetition of "the plates were incubated", as the subject is already clear from the previous sentence.
- corrected
Lines 133–134, should be revised similarly by removing redundant details while maintaining clarity.
- revised
The "Materials and Methods" section is well-documented; however, it could be slightly condensed by reformulating sentences to eliminate repetitions and redundant phrasing. The suggested revisions maintain clarity without losing essential information.
- We removed unnecessary repeats.
Results
For better reader comprehension, when presenting the results of Table 1, abbreviations should be used in the text. Additionally, the transitions before the analysis of isolates 7 and 8 should be improved. This would help the reader understand why these specific isolates are analyzed in detail and how their results relate to the data in the table.
- These two isolates were just samples as otherwise the other isolates were pretty homogenous, practically sensitive.
The section "Antimicrobial Effects of Essential Oils" should be revised to eliminate repetitions and to rephrase the results regarding the effect of essential oils. Additionally, you should provide information about the inhibition zones in the case of bitter orange essential oil.
- performed and conretized.
At the end of the "Biofilm Assay" section, after Figure 6, you presented the numerical results, but it is not clear why they are important and how they influence the treatment of the infection. You should establish a clearer connection between the experimental results and their clinical implications.
- practically it is a numerical description, -after figure 7 (now).- of the data, that might have been overemphasized. The two relevant sentences are deleted now. The remaining decsriptive part was left.
Discussion
Lines 432-434: You mentioned a low resistance rate compared to international data, but you did not provide an explanation for this. How do you explain this difference compared to similar studies? You should include a possible interpretation, considering factors such as the type of population studied, antibiotic usage, geographical variations, or applied methodologies.
- we do not know the exact reason of that but one option we have provided.
Lines 455-456: You should create a smoother transition between antibiotic use and the proposed solution (essential oils). Currently, the connection between these two aspects is not sufficiently clear, and a more detailed explanation could strengthen the justification for using essential oils as an alternative or adjunct in treatment.
- we partially reformulated this section that
Lines 460-463: You stated that out of the 57 essential oils tested, only 6 were effective, but you did not explain whether their effectiveness varied depending on concentration, exposure time, or specific conditions. It would be helpful to clarify these aspects to provide a more comprehensive interpretation of the results and facilitate comparisons with similar studies.
- we added some considerations with potential further
Lines 465-469: You stated that therapy is influenced by biofilm, but you did not explain how biofilm affects conventional treatments. You should provide a clear explanation of the mechanisms by which biofilm reduces antibiotic effectiveness, such as physical protection of bacteria, metabolic changes, or the development of antimicrobial tolerance and resistance. Additionally, it would be useful to emphasize why biofilm degradation is critical for treatment success.
- this issue is added now
Lines 480-482: You mentioned the antibacterial and anti-inflammatory effects of eugenol, but you did not explain how these properties could contribute to the treatment of external otitis. You should establish a clear connection between eugenol’s bioactive properties and its potential clinical role, explaining the mechanisms through which it may inhibit bacterial growth, reduce inflammation, and alleviate symptoms. This clarification would help in understanding how eugenol could be used as an adjunct or alternative treatment.
- Based on your suggestion and beeing aware that inflammation management is also important in otitis externa we tried to further demonstrate the role of Eugenol in this part and close Discussion with this issue. But we did not wanted to go in detail as this aspect is out of the scope of this study.
Conclusions
The conclusions should be reformulated to clarify the following aspects:
The necessity of antibiotic alternatives – Highlighting the increasing antimicrobial resistance and the urgent need to identify alternative solutions, such as essential oils. The relevance of this study should be emphasized in the context of finding sustainable and effective options.
The link between Pseudomonas aeruginosa, zoonoses, and the One-Health concept – Explaining how this bacterium impacts both human and veterinary health, thus justifying the importance of an integrated approach based on One-Health principles.
Feasibility of replacing antibiotics with essential oils – Mentioning the clinical and experimental context that supports the use of essential oils as alternatives or adjuncts in treatment. Additionally, discussing potential limitations and advantages, including compound stability, bioavailability, and administration methods.
The importance of the study and future research directions – Emphasizing the impact of this study in opening new research pathways regarding effective combinations of essential oils, their mechanisms of action, and applicability in biofilm-associated infections.
This clarification will strengthen the conclusions and provide a more scientifically grounded framework for the use of essential oils in combating P. aeruginosa-related infections.
In general, the references included in the manuscript are relevant and well-chosen for the context of the study. However, to enhance the scientific foundation and the up-to-date nature of the bibliography, I recommend adding recent studies (2020-2024) that address the use of essential oils in veterinary medicine and their effectiveness against Pseudomonas aeruginosa.
- thank You for your time, your commences and useful remarks! We tried to improve the quality of our study based on them.
Comments on the Quality of English Language
Sometimes, present and past tenses are incorrectly used within the same sentence.
Additionally, there are errors in article usage (a/an/the), and in some places, essential articles are missing.
Minor spelling mistakes are also present, (e.g. "infleuncing" instead of "influencing").
Some sentences are too long, leading to word order mistakes.
There are verb agreement errors and overly technical phrasing, lacking clear transitions between ideas.
Recommendation:
A professional proofreading or a careful review by a fluent English speaker would significantly improve clarity and readability.
Submission Date
12 February 2025
Date of this review
22 Feb 2025 23:27:24
Thank You for taking your time and contributing to the improvement of our manuscript!
György Schneider
corresponding author
Reviewer 2 Report
Comments and Suggestions for Authors
The submitted manuscript “Characterization of canine otitis externa Pseudomonas aeruginosa isolates and their sensitivities to different essential oils” focus on an interesting and relevant subject regarding monitorization of antimicrobials and replacements to these in veterinary medicine. Considering nowadays antimicrobial resistance, it is important to study other therapeutic alternatives to combat bacterial infections.
In general, the conceptualization of the study is correct.
The Simple Summary is satisfactory and well written. It provides information regarding the content of the manuscript.
The Abstract is correct and well written. It provides information regarding the content of the manuscript (objectives, materials and methods, results).
The keywords are adequate to the content of the study.
The introduction highlights the importance of the theme and provides correct known information regarding the subjects studied.
The methodology is well described and adequate to the objectives established.
The results are correctly exposed.
Line 208 – correct “…that all the of them were…” to: …that all of them were…
The Discussion is satisfactory.
Line 466- correct “infleuncing”
More than a reflexion, conclusions should be more objective and related to the results of the study.
Author Response
Reviewer 2
Comments and Suggestions for Authors
The submitted manuscript “Characterization of canine otitis externa Pseudomonas aeruginosa isolates and their sensitivities to different essential oils” focus on an interesting and relevant subject regarding monitorization of antimicrobials and replacements to these in veterinary medicine. Considering nowadays antimicrobial resistance, it is important to study other therapeutic alternatives to combat bacterial infections.
In general, the conceptualization of the study is correct.
The Simple Summary is satisfactory and well written. It provides information regarding the content of the manuscript.
The Abstract is correct and well written. It provides information regarding the content of the manuscript (objectives, materials and methods, results).
The keywords are adequate to the content of the study.
The introduction highlights the importance of the theme and provides correct known information regarding the subjects studied.
The methodology is well described and adequate to the objectives established.
The results are correctly exposed.
- Dear Reviewer, thank You for your general positive opinion about or study, also considering the below points like Summary, Abstract, Introduction, Methodology and Results!
Line 208 – correct “…that all the of them were…” to: …that all of them were…
- Thank You, corrected.
The Discussion is satisfactory.
Line 466- correct “infleuncing”
- corrected!
More than a reflexion, conclusions should be more objective and related to the results of the study.
- Based on your remark, we reformulated Conclusion in amore dircet and „future prospects” way.
György Schneider
corresponding author
Reviewer 3 Report
Comments and Suggestions for Authors
The study addresses an important issue—Pseudomonas aeruginosa infections in canine otitis externa and the potential of essential oils as alternative treatments. Given the increasing concern over antimicrobial resistance (AMR), this is a timely and relevant investigation, but some minor changes are required before publication.
Table 1 in the manuscript does not actually contain a balloon plot. Instead, it presents antibiotic susceptibility data in a numerical format, listing inhibition zone diameters for various antibiotics tested against Pseudomonas aeruginosa isolates. Including a smaller table summarizing the percentage of resistance to each antibiotic would be a much clearer and more effective way to present the data, rather than listing inhibition zone diameters for each isolate.
Discussion: Several sections in the discussion contain one-sentence paragraphs explaining resistance trends, essential oil effectiveness, or comparisons to previous studies. These could be combined into cohesive arguments, making the discussion more structured and engaging.
Figure 6 - I suggest increasing the size and resolution of Figure 6 to enhance readability. Currently, the figure appears too small, making it difficult to distinguish details, especially when comparing biofilm formation at 30°C and 37°C across different media and plate types. Enlarging the figure and ensuring high-resolution formatting will improve clarity and allow readers to interpret the data more effectively.
Conclusion
Include a statement about future studies, such as investigating the mechanism of action of essential oils, evaluating their synergistic effects with antibiotics, or conducting clinical trials in dogs.
Author Response
Reviewer 3
Comments and Suggestions for Authors
The study addresses an important issue—Pseudomonas aeruginosa infections in canine otitis externa and the potential of essential oils as alternative treatments. Given the increasing concern over antimicrobial resistance (AMR), this is a timely and relevant investigation, but some minor changes are required before publication.
- Thank You for your general positive opinion about our manuscript!
Table 1 in the manuscript does not actually contain a balloon plot. Instead, it presents antibiotic susceptibility data in a numerical format, listing inhibition zone diameters for various antibiotics tested against Pseudomonas aeruginosa isolates. Including a smaller table summarizing the percentage of resistance to each antibiotic would be a much clearer and more effective way to present the data, rather than listing inhibition zone diameters for each isolate.
- Your input is valuable and reasonable. From diagnostic point the table form is more informative with precise numbers that is why we have originally decided to leave only this table in the manuscript. From comparatve point of view the plot form is more reasonable as results of the EO inhibitions are represented in that way. For this reason the table (Table 1) was left and beside the plot diagram (Figure 1) of the antibiotic inhibition zone results were inserted.
Discussion: Several sections in the discussion contain one-sentence paragraphs explaining resistance trends, essential oil effectiveness, or comparisons to previous studies. These could be combined into cohesive arguments, making the discussion more structured and engaging.
- Thank you for your remarls. We thought about this part of the manuscript and modified it.
Figure 6 - I suggest increasing the size and resolution of Figure 6 to enhance readability. Currently, the figure appears too small, making it difficult to distinguish details, especially when comparing biofilm formation at 30°C and 37°C across different media and plate types. Enlarging the figure and ensuring high-resolution formatting will improve clarity and allow readers to interpret the data more effectively.
- Based on your remark we increased the size.
Conclusion
Include a statement about future studies, such as investigating the mechanism of action of essential oils, evaluating their synergistic effects with antibiotics, or conducting clinical trials in dogs.
- Based on your commence and commences of other reviewers we reformulated Conclusions.
Thank You again for your inputs concerning with our manuscript!
György Schneider
corresponding author
